# Newly Arrived Migrant Women’s Experience of Maternity Health Information: A Face-to-Face Questionnaire Study in Norway

**DOI:** 10.3390/ijerph18147523

**Published:** 2021-07-15

**Authors:** Sukhjeet Bains, Johanne Sundby, Benedikte V. Lindskog, Siri Vangen, Ingvil K. Sørbye

**Affiliations:** 1Norwegian Research Centre for Women’s Health, Department of Obstetrics and Gynecology, Oslo University Hospital, 0424 Oslo, Norway; sirvan@ous-hf.no (S.V.); isorbye@ous-hf.no (I.K.S.); 2Department of Community Medicine and Global Health, Institute of Health and Society, University of Oslo, 0316 Oslo, Norway; johanne.sundby@medisin.uio.no; 3Section for Diversity Studies, Department of International Studies and Interpreting, Oslo Metropolitan University, 0167 Oslo, Norway; benedik@oslomet.no; 4Institute of Clinical Medicine, Faculty of Medicine, University of Oslo, 0316 Oslo, Norway

**Keywords:** language barriers, health disparities, quality of care, migrants, maternity care, health literacy, interpreter, maternal health

## Abstract

Limited understanding of health information may contribute to an increased risk of adverse maternal outcomes among migrant women. We explored factors associated with migrant women’s understanding of the information provided by maternity staff, and determined which maternal health topics the women had received insufficient coverage of. We included 401 newly migrated women (≤5 years) who gave birth in Oslo, excluding migrants born in high-income countries. Using a modified version of the Migrant Friendly Maternity Care Questionnaire, we face-to-face interviewed the women postnatally. The risk of poor understanding of the information provided by maternity staff was assessed in logistic regression models, presented as adjusted odds ratios (aORs), with 95% confidence intervals (CI). The majority of the 401 women were born in European and Central Asian regions, followed by South Asia and North Africa/the Middle East. One-third (33.4%) reported a poor understanding of the information given to them. Low Norwegian language proficiency, refugee status, no completed education, unemployment, and reported interpreter need were associated with poor understanding. Refugee status (aOR 2.23, 95% CI 1.01–4.91), as well as a reported interpreter need, were independently associated with poor understanding. Women who needed but did not get a professional interpreter were at the highest risk (aOR 2.83, 95% CI 1.59–5.02). Family planning, infant formula feeding, and postpartum mood changes were reported as the most frequent insufficiently covered topics. To achieve optimal understanding, increased awareness of the needs of a growing, linguistically diverse population, and the benefits of interpretation services in health service policies and among healthcare workers, are needed.

## 1. Introduction

Due to increasing international migration, healthcare workers in host countries are providing care to an increasingly linguistically and culturally diverse patient group. Newly arrived migrants constitute a vulnerable group who, in addition to the loss of social status, discrimination, and socioeconomic marginalisation, may experience language barriers [1]. An increasing and considerable proportion of women giving birth in host countries are migrants. Thus, maternity care is often among the first exposures to a new healthcare system for migrant women. In addition, pregnancy and birth may exacerbate already existing vulnerability factors.

Disparities in maternal health outcomes and sub-optimal healthcare for migrants in Europe have been well documented [2,3]. Migrants have poorer access to, and inadequate utilisation of, available maternity healthcare services, which may be associated with socioeconomic status and the reason for migration [4]. Furthermore, women born in low- or middle-income countries represent a group with a higher risk-profile and in need of healthcare during pregnancy and birth [5,6]. While the causes of disparities are multifactorial, inadequate uptake of maternity health information and the ability to act on this information has been suggested as a major contributor, particularly for newly arrived migrants [7,8].

Adequate understanding of health information provided face-to-face by a health worker depends on several factors, such as health literacy, language proficiency, and the cultural competence and communication skills of both the patient and healthcare worker [9,10,11,12]. In addition, migrant background, educational level, and occupational and economic status can also influence the understanding of the health information of a patient [13,14,15].

The use of a professional interpreter has been shown to reduce the language barrier and improve the quality of care [16,17,18]. Provision of interpretation services is furthermore a modifiable factor that may be handled from within the healthcare system, in contrast to more complex factors such as socioeconomic status. Consequently, a number of European countries aim to provide interpreter services to migrants [19].

We know that the health information need is particularly high during pregnancy and birth, due to significant physical and psychological changes, in addition to the concerns about the foetus [20]. Moreover, the health information need is critical, as behaviours can have long-term consequences for women and their offspring [21]. Poor understanding can impact timely access to maternity care services, and impact the patient–provider relationship [22]. Ultimately, it may lead to poor compliance, and in the worst case, adverse outcomes [23,24].

Currently, little is known about newly arrived migrant women’s experiences of receiving, and level of understanding, health information in maternity care. In this study, we conducted face-to-face structured interviews with newly arrived migrant women in Norway, a country where almost 30% of the children born in 2020 had a migrant mother [25]. We explored factors associated with newly arrived migrants’ understanding of information provided by maternity staff. In addition, we determined which maternal health topics the women had received insufficient coverage of.

## 2. Materials and Methods

### 2.1. Study Design

This questionnaire study is a part of the larger MiPreg-project that is seeking to identify factors that explain disparities in maternity outcomes among newly migrated women in urban Oslo, Norway. The project is a multidisciplinary, mixed-method project with qualitative and quantitative work packages. For this quantitative study, we used a modified version of the Migrant Friendly Maternity Care Questionnaire (Appendix A). This structured questionnaire on maternity care was developed to be used in migrant populations [26]. It includes information on maternal socioeconomic factors, migration and obstetric characteristics, and understanding of information and interpreter use. The original questionnaire was adapted to the health system setting of Norway and modified to include questions on socio-economic background from national surveys. Response options for questions about antenatal services used by the women were altered to fit current available services within the healthcare system in Norway. Furthermore, we conducted pilot-testing of the questionnaire and made adjustments accordingly.

### 2.2. Study Setting

Norway has universal health coverage, and essential healthcare before, during, and after birth is free of charge for all legal citizens. Persons without legal residence have the right to healthcare, and if they cannot pay for maternity services they are exempted [27]. The standard antenatal package offered to low-risk pregnancies, with eight consultations, includes one routine ultrasound examination around week 18. Antenatal care is provided by a general practitioner or midwife in low-risk pregnancies, and by obstetricians in high-risk pregnancies. Patients have a legal right to receive healthcare information in a language they understand, free of charge [28]. It is the responsibility of the healthcare worker to book an interpreter, and it is recommended that relatives should not be used in place of a professional interpreter [28].

### 2.3. Study Population

We included international migrant women who gave birth in urban Oslo, with a length of stay in Norway ≤ 5 years. We excluded migrant women born in high-income countries, as defined by the Global Burden of Disease framework, which is based on epidemiological similarity and geographic closeness [29]. The woman’s country of birth was further classified into the Global Burden of Disease super-regions; Latin America & the Caribbean, Sub-Saharan Africa, North Africa & the Middle East, South East Asia, East Asia & Oceania, South Asia and Central Europe, Eastern Europe & Central Asia.

### 2.4. Data Collection

From January 2019 to January 2020, eligible women were recruited by trained research personnel, a physician, and three midwives from the two public hospitals with a maternity ward that serve urban Oslo (approximately 14,800 births annually): Oslo University Hospital and Akershus University Hospital. Almost all births in Norway are institutionalised and occur in public hospitals. The research personnel went through the maternity ward list approximately once a week and identified eligible women by asking the midwife in charge about the women’s country of birth and length of stay in Norway. As such, eligible participants were women admitted to the ward the days we recruited participants, i.e., consecutive selection was used. If eligible, written consent was obtained after informing the women about the study, using an interpreter if needed. The research personnel conducted the interviews face-to-face with the women at the postnatal ward 1–3 days after birth, in the woman’s language of choice, using an interpreter, when needed. Training workshops for the research personnel were conducted, and an interview guidebook was produced to ensure accuracy and consistency in registration.

### 2.5. Outcome Variables

We explored the women’s understanding of information by asking the question “*Did you understand the information the health care worker tried to convey to you*?” combined for three time periods; during pregnancy, during birth, and after birth. As the distribution of the response data was strongly skewed towards *always understood*, we categorised the data as a binary variable: *good understanding*, which included “*always understood the information*”, and *poor understanding*, which included “*sometimes*”, “*rarely*” and “*never understood the information*”. Further, the women were asked to determine whether they had received sufficient or insufficient coverage of a range of maternal health topics during the course of their pregnancy.

### 2.6. Explanatory Variables

We determined majority language proficiency by asking about the level of Norwegian fluency for oral, reading, writing, and comprehension skills, with the response options “*fluent*”, “*good*”, “*some difficulty*”, and “*not at all*”. A sum-score ranging from 4 to 16 was created, and we grouped the variable into tertiles; “*Low*” with a sum-score of 4–7; “*Moderate*” with a sum-score of 8–11; and “*High*”, with a sum-score of 12–16. As to the reason for migration, we used the national classification based on the legal grounds for immigration, grouping women into three categories: refugee, work/education, and family reunification. Completed maternal education was classified into three groups: no education, primary and secondary school, or university. The need for and offer of a professional interpreter was assessed for the three time periods: during pregnancy, during birth, and after birth.

### 2.7. Statistical Analysis

Descriptive statistics, such as the means with standard deviations (SD) and frequencies with percentages, were calculated for categorical and continuous variables. There were no missing values. To test differences between poor and good understanding, we used chi-square tests for all categorical variables, and Mann-Whitney Tests for the continuous variables. Associations between explanatory variables and poor understanding were estimated by univariable and multivariable logistic regressions, presented as crude (OR) and adjusted odds ratios (aOR) with 95% confidence intervals (CI). In Model A, we adjusted for majority language proficiency, the reason for migration, education, and employment. In Model B, we additionally included the variable offered interpreter during pregnancy. We only included the time period of pregnancy as it comprised the period where most women reported a need for a professional interpreter. In addition, we explored a possible interaction effect between majority language proficiency and if the woman had been offered an interpreter during pregnancy. However, as the interaction term was not significant in the model, we excluded it. We assessed the goodness of fit for the regression models and checked for multicollinearity. The significance level was set at 0.05. The analyses were performed with IBM SPSS version 25.

### 2.8. Ethics and Public Involvement

This study was approved by each hospital’s Ethical Review Committee (approval 18/15786 + 18/05310). Written informed consent was obtained from the women who participated in the study. User representatives from migrant communities were involved from the design phase, and throughout the implementation phase, of the MiPreg study.

## 3. Results

### 3.1. Characteristics

A total of 401 newly migrated women, born in 65 different countries, were interviewed (87.5% response rate). Overall, one-third (33.4%) of the women reported a poor understanding of the information provided by maternity staff during their pregnancy, birth, or after birth. The majority of women were born in the Central/Eastern European and Central Asian regions, followed by South Asia and North Africa/the Middle East. As to the women’s country of birth, the top five represented countries were Poland (10.2%), Pakistan (8.1%), India (7.7%), the Philippines (6.5%), and Eritrea (5.5%). The mean age was 29.8 years, and the mean length of residency was 36 months. Understanding of information did not differ significantly between primiparous and multiparous women. Among women reporting a poor understanding, most had a low majority language proficiency, while among women reporting a good understanding, most had high proficiency. Overall, more than half had a university education, and almost 60% were employed. More women without any completed education reported poor understanding (56.2%), while the majority of the women with a completed university degree reported good understanding (70.9%). Overall, the majority had migrated due to family reunification or work/education, while 10.2% were refugees. More refugees reported poor understanding (51.2%), while more women who migrated due to education/work reported a good understanding (72.9%). The women’s need for an interpreter varied during the three time periods, with the highest need reported during pregnancy (42.1%). Among those who felt the need for an interpreter, most of them were offered one during pregnancy (56.2%), whereas few women were offered one during birth (19.0%) (Table 1).

The baseline characteristics varied between the women who needed but did not get an interpreter, those who needed and did get an interpreter, and those who did not need a professional interpreter (Appendix A). Women with refugee status were offered a professional interpreter during pregnancy, birth, and after birth most often (41.5%, 9.8%, and 29.3%, respectively). Partners or other adult family members were most commonly used as interpreters (74.0%), followed by a professional interpreter (19.2%) or a bilingual healthcare worker (5.1%). Only one woman reported that her underage child had been used as an interpreter (data not shown).

### 3.2. Factors Associated with Poor Understanding of Information

The majority language proficiency, reason for migration, educational level, employment, and offer of a professional interpreter during pregnancy were associated with poor understanding of information in the crude analysis (Table 2). Needing but not getting offered a professional interpreter during pregnancy increased the risk of poor understanding of information (crude OR 3.30, 95% CI 1.91–5.70). In model A, women with low majority language proficiency (aOR 2.14, 95% CI 1.14–4.02) were more likely to have a poor understanding of information compared to those with high proficiency (Table 2). Furthermore, women who migrated as refugees (aOR 2.56, 95% CI 1.18–5.53, Table 2) had a higher risk of poor understanding compared to women who migrated due to education or work. In model B, the reason for migration and being offered a professional interpreter during pregnancy remained statistically significant (Table 2). The women who needed, but did not get offered, a professional interpreter were 2.8 times more likely to have a poor understanding of information, whereas those who needed and got one were 2.1 times more likely to have a poor understanding of information, compared to those who did not need a professional interpreter.

### 3.3. Insufficient Coverage of Maternal Health Topics

More than half of the women reported insufficient coverage on the topic of family planning (58%), infant formula feeding (56%), and postpartum mood changes (53%). Information about recommended medical tests had the lowest reported proportion of insufficient coverage (17%). For all topics, higher proportions of insufficient coverage were reported by the women with a poor understanding of information, compared to women with a good understanding (Figure 1).

## 4. Discussion

Among 401 newly arrived migrants, one-third (33.4%) reported a poor understanding of information provided by maternity staff during pregnancy, birth, or after birth. Needing, but not getting offered, a professional interpreter during pregnancy, compared to not needing one, increased the risk of poor understanding (aOR 2.83, 95% CI 1.59–5.02). In addition, refugee status, compared to having migrated due to education or work, also increased the risk of poor understanding (aOR 2.23, 95% CI 1.01–4.91). More than half of the women reported insufficient coverage of family planning, infant formula feeding, and postpartum mood changes.

### 4.1. Poor Understanding of Information

Migrant women’s poor understanding of the information provided by maternity staff has been well documented in qualitative studies [30,31]. We show that being offered a professional interpreter was associated with a better understanding of information. We also found an unmet need for professional interpreter services, consistent with the literature [30,32,33,34]. Thus, these results suggest that more effort should be put into providing interpreting services, which compared to other factors, is a more easily modifiable factor. This is in line with a WHO report which identified interpretation, translation, cultural mediation, and education of healthcare workers as the most significant strategies for reducing communication barriers for migrants in Europe [35].

However, several factors can cause the underuse of interpreting services. A Swiss study reported that only 9% of healthcare workers had received training in the importance of, and how to work with, a professional interpreter [36]. In addition, very few healthcare workers expressed that their health facility encouraged using professional interpreters [36]. Increased awareness among policymakers, as well as continued education for healthcare workers about their responsibility to provide measures for better understanding, were indicated as important in a previous Norwegian study [32]. Targeted actions to increase the use of professional interpreters for women during birth has shown positive results [37]. Additionally, interventions designed to increase understanding of information among patients with low health literacy, such as adding video to written information or pictograms, has led to improved comprehension [38].

As expected, the offer of a professional interpreter was most common during antenatal care, possibly due to the structure of the consultations, with a set time frame and therefore easier logistics. Although ensuring a good understanding of information is crucial during birth, not only to avoid adverse maternal outcomes such as perineal tears but also for the birth experience of the woman, only 19% of the women who needed interpretation were offered it. Our findings, therefore, indicate that the recommended standards for providing patients with interpretation services in Norway are not being followed. This was also found in an Australian study, which reported that only 22% of the women who did not speak English had access to a professional interpreter during birth [39].

In contrast to countries with considerable linguistic diversity among maternity staff, such as the UK, bilingual maternity staff were seldom used as interpreters in our study [33]. This emphasises the need for other strategies to overcome language barriers in countries with less linguistic diversity among healthcare workers. Consistent with our findings, using family members as interpreters was a common strategy to overcome language barriers; however, this is not recommended, or in accordance with guidelines [33,34,40].

Our findings of a poor understanding of information among refugees may partially explain insufficient access and utilisation of antenatal care within this subgroup of migrants [4,41]. The majority language proficiency is undoubtedly an important factor in understanding information, as confirmed by other studies [42,43]. However, it can only partially explain differences, as a substantial proportion of women with low and moderate language proficiency reported adequate understanding. It is worth mentioning that our findings do not take into account whether or not the women spoke English, a language many healthcare workers in Norway have a good command of. Therefore, women with low to moderate Norwegian proficiency with good understanding might represent those who spoke English. In agreement with our study, parity has been shown to not be associated with the level of understanding of health information [44].

### 4.2. Insufficient Coverage of Maternal Health Topics

We found a high rate of insufficient coverage of several maternal health topics. Among women who reported poor understanding of information, a greater proportion of topics were reported to be insufficiently covered. In line with our findings of insufficient coverage about family planning, a German study found that although the government provided free family planning services, there was a big knowledge gap for refugees [45]. Interventions with the aim of increasing knowledge about family planning may be particularly important for migrants, as some originate from countries with minimal sexual and reproductive education in school. Infant formula feeding was the second most frequent topic with insufficient coverage. In Norway, exclusive breastfeeding is recommended for the first six months and, if possible, throughout the first year of life, and preferably longer. Although breastfeeding is more common among women in low- or middle-income countries [46], migration to a high-income country generally tends to have a negative impact on breastfeeding practices [47,48]. Maternity staff may therefore be hesitant to provide information on infant formula feeding, as they may fear that it leads to its overuse. A systematic review concluded that the high use of early supplementation with formula among African migrants was due to the belief that formula is necessary to achieve bigger, and thus healthier, babies [49]. Better education about indications, benefits, and disadvantages regarding infant formula feeding is needed. The women in our study also reported high rates of insufficient coverage of changes in mood postpartum. Higher rates of perinatal depression among migrants have been found previously [50]. As insufficient information and stigma about depression has an impact on help-seeking behaviour [51], ensuring better education about symptoms and the importance of seeking help in time is crucial.

### 4.3. Strengths and Limitations

A strength of this study was the use of extensive face-to-face interviews, with interpretation provided as needed. This enabled all women to participate, not excluding illiterate women or limiting inclusion to certain languages. As such, it reduced the chance of selection bias and missing data, as well as information bias due to misinterpretation of questions. We had a high response rate of 87.5%, and the non-participating women did not differ from the participants in terms of age, length of residence in Norway, or region of birth.

Nevertheless, some limitations to our study should be addressed. First, the questionnaire was administered shortly after birth to ensure responses from hard-to-reach groups, as postpartum care is fragmented in Norway. However, as new mothers may be tired and might have a hard time remembering details about the pregnancy at this time, this might have impacted the answers. Second, social desirability bias, where the women over-report “good behaviour” and socially acceptable answers, may have affected our questionnaire since the interviews were held at the ward. However, the research staff did not partake in clinical patient care, which was carefully explained at recruitment. Third, not including a variable measuring English proficiency most likely limited our interpretation of language proficiency regarding the understanding of information. As English-speaking women may report good understanding despite having low Norwegian proficiency, the language variable may in reality be more strongly associated with understanding than what can be seen from our findings. Furthermore, as the consecutive selection was applied, the findings apply primarily to newly arrived migrants in urban Oslo. Due to heterogeneity in the composition of migrant women in different countries, caution must be taken when generalizing the results.

## 5. Conclusions

Our study contributes to the identification of modifiable factors that could improve newly arrived migrant women’s understanding of maternity health information, as well as identifying gaps in the coverage of maternal health topics. Our findings of suboptimal provision of interpreting services, alongside an improved understanding among women who did get offered a professional interpreter, suggest that current policies are yet to be put into consistent practice. Targeted interventions should be applied to adapt healthcare services to linguistically diverse patients, including the provision of tailored health education and prenatal classes that consider the specific needs of newly arrived migrants.

## Figures and Tables

**Figure 1 ijerph-18-07523-f001:**
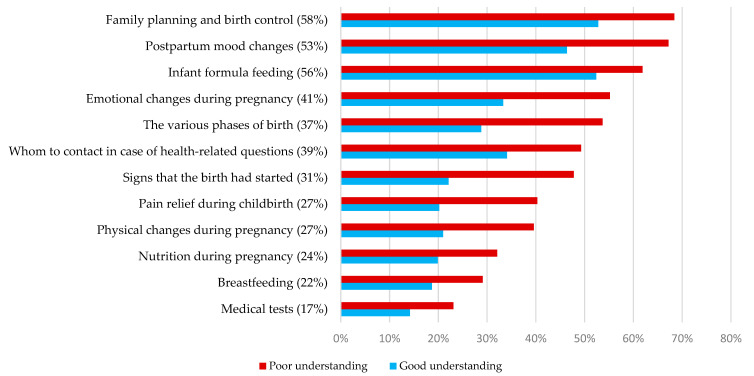
Proportion of women who reported receiving insufficient coverage of various maternal health topics (percentage of all) and by poor and good understanding of information provided by maternity staff.

**Table 1 ijerph-18-07523-t001:** Characteristics of all study participants and according to poor or good understanding of information provided by maternity staff, n (%) or mean (SD).

Characteristics	All Participants (N = 401)	Poor Understanding (N = 134)	Good Understanding (N = 267)	*p*-Value
Mean age, in years (SD)	29.8 (4.7)	29.4 (4.5)	30.0 (4.8)	0.188 ^a^
Mean length of residency, in months (SD)	35.6 (19.4)	32.9 (18.6)	37.0 (19.7)	0.044 ^a^
Women region of birth (global burden of disease), n (%)				0.067 ^b^
Central/Eastern Europe and Central Asia	132 (32.9)	37 (28.0)	95 (72.0)	
Latin America and the Caribbean	13 (3.2)	3 (23.1)	10 (76.9)	
North Africa and the Middle East	76 (19.0)	29 (38.2)	47 (61.8)	
South Asia	81 (20.2)	23 (28.4)	58 (71.6)	
Southeast Asia, East Asia and Oceania	37 (9.2)	12 (32.4)	25 (67.6)	
Sub-Saharan Africa	62 (15.5)	30 (48.4)	32 (51.6)	
Partner’s background, n (%)				0.061 ^b^
Norwegian	54 (13.5)	12 (22.2)	42 (77.8)	
Foreign	347 (86.5)	122 (35.2)	225 (64.8)	
Parity, n (%)				0.919 ^b^
Primiparous	229 (57.1)	77 (57.5)	152 (56.9)	
Multiparous	172 (42.9)	57 (42.5)	115 (43.1)	
Majority language proficiency, n (%)				0.017 ^b^
Low	112 (27.9)	47 (42.0)	65 (58.0)	
Moderate	173 (43.1)	59 (34.1)	114 (65.9)	
High	116 (28.9)	28 (24.1)	88 (75.9)	
Education, n (%)				0.030 ^b^
No completed school	16 (4.0)	9 (56.2)	7 (43.8)	
Primary/secondary school	151 (37.7)	57 (37.7)	94 (62.3)	
University	234 (58.4)	68 (29.1)	166 (70.9)	
Employment, n (%)				0.017 ^b^
Unemployed	173 (43.1)	69 (39.9)	104 (60.1)	
Employed	228 (56.9)	65 (28.5)	163 (71.5)	
Financial level, n (%)				0.028 ^b^
High	313 (78.1)	96 (30.7)	217 (69.3)	
Low	88 (21.9)	38 (43.2)	50 (56.8)	
Reason for migration, n (%)				0.009 ^b^
Refugee	41 (10.2)	21 (51.2)	20 (48.8)	
Family reunification	183 (45.6)	65 (35.5)	118 (64.5)	
Education/work	177 (44.1)	48 (27.1)	129 (72.9)	
Need for and offer of a professional interpreter during pregnancy, n (%)				<0.0001 ^b^
Needed but did not get	74 (18.5)	37 (50.0)	37 (50.0)	
Needed and got	95 (23.7)	43 (45.3)	52 (54.7)	
Did not need	232 (57.9)	54 (23.3)	178 (76.7)	
Need for and offer of a professional interpreter during birth, n (%)				<0.0001 ^b^
Needed but did not get	128 (31.9)	63 (49.2)	65 (50.8)	
Needed and got	30 (7.5)	15 (50.0)	15 (50.0)	
Did not need	243 (60.6)	56 (23.0)	187 (77.0)	
Need for and offer of a professional interpreter after birth, n (%)				
Needed but did not get	102 (25.4)	45 (44.1)	57 (55.9)	<0.0001 ^b^
Needed and got	54 (13.5)	33 (61.1)	21 (38.9)	
Did not need	245 (61.1)	56 (22.9)	189 (77.1)	

^a^ Mann-Whitney Test (2-tailed). ^b^ Pearson Chi-Square (2-sided).

**Table 2 ijerph-18-07523-t002:** Factors associated with poor understanding of information given by healthcare personnel during pregnancy, birth, and after birth.

Factors	Crude OR (95% CI)	Adjusted OR (95% CI) Model A	Adjusted OR (95% CI) Model B
Majority language proficiency			
Low	2.27 (1.29–4.01) *	2.14 (1.14–4.02) *	1.76 (0.92–3.40)
Moderate	1.63 (0.96–2.76)	1.51 (0.87–2.62)	1.26 (0.71–2.23)
High	1.00	1.00	1.00
Reason for migration			
Refugee	2.82 (1.41–5.66) *	2.56 (1.18–5.53) *	2.23 (1.01–4.91) *
Family reunification	1.48 (0.95–2.32)	1.40 (0.85–2.31)	1.37 (0.82–2.27)
Education/work	1.00	1.00	1.00
Education			
No completed school	3.14 (1.12–8.77) *	1.78 (0.60–5.29)	1.26 (0.41–3.86)
Primary/secondary school	1.48 (0.96–2.28)	1.13 (0.71–1.81)	0.93 (0.56–1.54)
University	1.00	1.00	1.00
Employment			
Unemployed	1.66 (1.10–2.53) *	1.16 (0.72–1.87)	1.05 (0.63–1.73)
Employed	1.00	1.00	1.00
Need for and offer of a professional interpreter during pregnancy			
Needed but did not get	3.30 (1.91–5.70) *		2.83 (1.59–5.02) *
Needed and got	2.73 (1.64–4.52) *		2.07 (1.14–3.76) *
Did not need	1.00		1.00

* Significant at *p* < 0.05. OR = Odds ratio. CI = confidence interval. GBD = global burden of disease. Model A: includes “*majority language proficiency*”, “*reason for migration*”, “*education*” and “*employment*”. Model B: includes model A + “*offered professional interpreter during pregnancy*”.

## Data Availability

The datasets generated and analysed during the current study are not publicly available due to protection of individual participants’ privacy and confidentiality, but are available from the corresponding author on reasonable request.

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
