# Peer review of "Newly Arrived Migrant Women’s Experience of Maternity Health Information: A Face-to-Face Questionnaire Study in Norway"

_ijerph, 2021, doi:10.3390/ijerph18147523_

Round 1
Reviewer 1 Report
Method needs more clarity. Give more details of the mixed method, for example. Describe the modified questionnaire surveys.
Author Response
Thank you for reviewing our manuscript and for highlighting the need for more details about methods.
We have now altered the information about the MiPreg project, see page 2, line 81-86:
“This questionnaire study is a part of the larger MiPreg-project that is seeking to identify factors that explain disparities in maternity outcomes among newly migrated women in urban Oslo, Norway. The project is a multidisciplinary, mixed method project with qualitative and quantitative work packages. For this quantitative study, we used a modified version of the Migrant Friendly Maternity Care Questionnaire (Supplementary 1)”.
We have also edited the sentence about the modified questionnaire and explained in more detail what modifications were done, see page 2, line 89-93:
“The original questionnaire was adapted to the health system setting of Norway, and modified to include questions on socio-economic background from national surveys. Response options for questions about antenatal services used by the women were altered to fit current available services within the healthcare system in Norway. Furthermore, we conducted pilot-testing of the questionnaire, and made adjustments accordingly”.
We have also added more details on how the participants were recruited, see page 3, line 120-125:
“Almost all births in Norway are institutionalised and occur in public hospitals. The research personnel went through the maternity ward list approximately once a week and identified eligible women by asking the midwife in charge about the women’s country of birth and length of stay in Norway. As such, eligible participants were women admitted to the ward the days we recruited participants, i.e., consecutive selection was used “.
We hope we have sufficiently informed about the MiPreg project and modification of the questionnaire in methods.
Reviewer 2 Report
Thank you for the opportunity to review the manuscript “Newly Arrived Migrant Women’s Experience of Maternity Health Information – a Face-to-face Questionnaire Study in Norway” for International Journal of Environmental Research and Public Health. Thank you for conducting such an important study. Overall, it was clearly written and thought provoking. I do, however, have a suggestion on how the manuscript could be improved:
The authors state that this was a cross-sectional study. However, the authors do not provide any information on how participants recruited “from two public hospitals that serve 110 urban Oslo” constitute a representative cross section of newly migrated women who gave birth in Oslo. My suspicion is that the authors relied on nonprobability methods to gather these data. This is not a small issue I am raising. If my suspicion is correct, the findings cannot be generalized beyond the sample. There is nothing inherently wrong with using a nonprobability approach, especially considering this is an exploratory study (which, by the way, should be stated in the methods section), however it is factually incorrect to refer to any nonprobability as a cross-sectional sample. In my view the scholarship of this work would be strengthened considerably if the author(s) would be more explicit in their description of the data collection efforts.
Author Response
Thank you for reviewing our manuscript and for your relevant comments. The English used in this manuscript has been checked and edited via English Editing Services from MDPI.
We agree with the reviewer that the recruitment process needs more clarity. We have removed the description “cross-sectional” from the manuscript. We have also added more details on how the participants were recruited, see page 3, line 117-125:
“From January 2019 to January 2020, eligible women were recruited by trained research personnel, a physician, and three midwives from the two public hospitals with a maternity ward that serve urban Oslo (approximately 14,800 births annually): Oslo University Hospital and Akershus University Hospital. Almost all births in Norway are institutionalised and occur in public hospitals. The research personnel went through the maternity ward list approximately once a week and identified eligible women by asking the midwife in charge about the women’s country of birth and length of stay in Norway. As such, eligible participants were women admitted to the ward the days we recruited participants, i.e., consecutive selection was used “.
We recruited women from the two maternity wards serving urban Oslo. Since almost all births finds place in hospitals in Norway (1) we believe we were able to identify almost all eligible women for this study. We were not able to recruit women every day and have therefore most likely missed some women who were discharged after giving birth, before we could inform them about the study. However, we do not have reason to believe that the recruited women and those discharged before we were able to recruit differed significantly.
We thank the reviewer for highlighting the problems with generalizability; understanding of information is a complex outcome that may differ in different groups depending on several factors. We do, however, believe that the findings in this study are relevant to European countries receiving migrants from the same countries of origin. The following text has been included at the end of limitations, see page 10, line 339-342:
“Furthermore, as consecutive selection was applied, the findings apply primarily to newly arrived migrants in urban Oslo. Due to heterogeneity in the composition of migrant women in different countries, caution must be taken when generalizing the results.”
We hope we have sufficiently informed about the recruitment process and why we believe it is representative of newly migrated women giving birth in Oslo.
Reference:
- The Norwegian Government: Norske kvinner føder på sykehus (Norwegian women give birth at hospitals). Accessed from: https://www.regjeringen.no/no/aktuelt/norske-kvinner-foder-pa-sykehus/id2666411/